# FedID: Federated Interactive Distillation for Large-Scale Pretraining Language Models

**Xinge Ma, Jiangming Liu**[*]**, Jin Wang**[*] **and Xuejie Zhang**
School of Information Science and Engineering
Yunnan University
Kunming, China
Contact:jiangmingliu@ynu.edu.cn, wangjin@ynu.edu.cn

## Abstract

The growing concerns and regulations surrounding the protection of user data privacy have necessitated decentralized training paradigms. To this end, federated learning (FL) is widely studied in user-related natural language processing (NLP). However, it suffers from several critical limitations including extensive communication overhead, inability to handle heterogeneity, and vulnerability to white-box inference attacks. Federated distillation (FD) is proposed to alleviate these limitations, but its performance is faded by *confirmation bias*. To tackle this issue, we propose Federated Interactive Distillation (FedID), which utilizes a small amount of labeled data retained by the server to further rectify the local models during knowledge transfer. Additionally, based on the GLUE benchmark, we develop a benchmarking framework across multiple tasks with diverse data distributions to contribute to the research of FD in NLP community. Experiments show that our proposed FedID framework achieves the best results in homogeneous and heterogeneous federated scenarios. The code for this paper is available at: https://github.com/maxinge8698/FedID.

## 1 Introduction

The remarkable success of natural language processing (NLP) is highly dependent on large-scale pre-trained language models (PLMs; Devlin et al. 2019; Liu et al. 2019; Yang et al. 2019; Clark et al. 2020). To fully realize the potential of PLMs, they are typically trained using large amounts of combined data that is collected from multiple distributed user devices (*a.k.a.*, clients) and transmitted to a single data center (*a.k.a.*, server). With the growing concerns about privacy protection, data regulations such as the Personal Data Protection Act (PDPA; Chik 2013) and the General Data Protection Regulation (GDPR; Voigt and von dem

---
[*]Corresponding authors.

Bussche 2017) have imposed strict requirements on preserving user data privacy, making it impractical to aggregate such data to a centralized location for training. Federated learning (FL; Mcmahan et al. 2017) has emerged as a privacy-preserving decentralized training paradigm, in which a federation of clients is orchestrated by a central server to collaboratively train a shared global model via aggregating the local models trained on their respective data. As a result, the private data of massive clients is effectively exploited in the form of model parameter exchange to train a unified model for better performance than individually working.

Previous work on federated NLP mainly targets solving either word-level language modeling applications such as mobile keyboard suggestion (Ji et al., 2019) and recommendation (Lin et al., 2020), or biomedical named entity recognition (Liu and Miller, 2020; Ge et al., 2020; Sui et al., 2020). More recently, Lin et al. (2022) provide a research-oriented benchmarking framework for advancing FL in NLP. However, these federated NLP frameworks are limited to identical architectures across the server and clients, making it impossible for clients to design their models independently according to their inconsistent system resources and non-independent and identically distributed (non-IID) data. Also, the frequent model parameter exchange entails expensive communication costs. These obstacles significantly hinder the applicability and scalability of FL for large-scale PLMs.

Instead, federated distillation (FD) eliminates the need to share model parameters by transferring knowledge from the clients to the server using an unlabeled public proxy dataset (Jeong et al., 2018; Li and Wang, 2019; Chang et al., 2019; Gong et al., 2022; Itahara et al., 2021; Hu et al., 2021), thereby allowing collaboration between heterogeneous models with less communication costs. However, FD suffers from confirmation bias (Arazo et al., 2020; Pham et al., 2021) induced by incor-

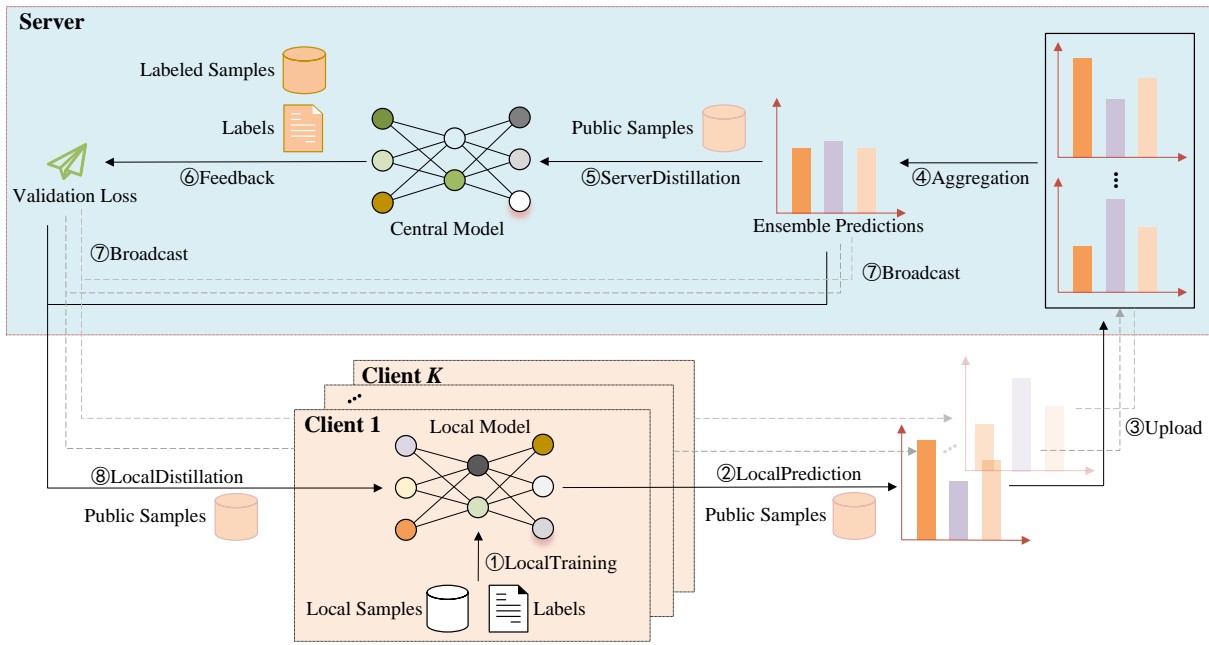

Figure 1: The framework of FedID.

porating incorrect or even biased predictions on unlabeled data for knowledge transfer, where the central model will tend to degenerate.

To tackle this challenge, we propose Federated Interactive Distillation (FedID), where a small handful of labeled data is retained in the server, aiming to provide feedback to the local models to debias the predictions.

In addition, previous studies on FD tend to design different partitioning strategies on different datasets, and only small-scale models are taken into consideration, which makes it difficult to evaluate and compare various FD approaches scaled to the NLP domain in a systematic and fair manner. For this reason, based on the General Language Understanding Evaluation (GLUE) benchmark (Wang et al., 2019), we create a unified benchmarking framework across multiple tasks with diverse data distributions to simulate a variety of federated scenarios for evaluating the effectiveness of these methods on the decentralized training of large-scale PLMs, advancing the research of FD in NLP. Empirical experiments show that our proposed FedID achieves the best results in homogeneous and heterogeneous federated scenarios.

The contributions of this paper are summarized as follows:

- To the best of our knowledge, we are the first to investigate the application of FD to decentralized learning of large-scale PLMs in ho-

mogeneous and heterogeneous settings.

- We present a novel Federated Interactive Distillation framework to mitigate the problem of misleading privileged knowledge caused by confirmation bias in conventional FD.

- We provide a unified benchmarking framework across multiple NLP tasks with diverse data distributions to contribute to the research of FD in NLP community.

## 2 Related Work

### 2.1 Federated Learning

FL has gained significant interest and attention in the NLP field due to its potential for collaborative training on distributed data sources while preserving data privacy (Liu et al., 2021). Recent efforts have made preliminary explorations for the application of parameter averaging-based FL (*e.g.*, FedAvg (Mcmahan et al., 2017)) in the context of NLP (Tian et al., 2022; Dong et al., 2022; Zhang et al., 2022; Lin et al., 2022). Despite some success, several system-oriented challenges have to be faced to make FL widely available in NLP, including extensive communication overhead, inability to handle heterogeneity, and vulnerability to white-box inference attacks.

Several variants of FL have emerged to attempt to alleviate these issues. FedDF (Lin et al.) builds prototypical models with the same structure as the

client models on the server side to enable model heterogeneity, and allows server-side ensemble distillation on unlabeled data from other domains to enhance model aggregation. FedED (Sui et al., 2020) reduces uplink communication costs by uploading the predictions of the local models instead of the parameters to train the central model, but still requires broadcasting the parameters of the central model over the downlink. Accordingly, these solutions still rely on exchanging model parameters and therefore are unable to completely address these limitations.

## 2.2 Federated Distillation

FD is a new algorithmic paradigm for FL with fundamentally different communication properties by exchanging the knowledge obtained during the local training in the form of model outputs rather than model parameters. This shared knowledge can be an aggregated statistic of model outputs on local private data (Jeong et al., 2018) or an ensemble of local model outputs computed on a publicly available proxy dataset (Li and Wang, 2019; Chang et al., 2019; Gong et al., 2022; Itahara et al., 2021; Hu et al., 2021). Existing efforts on FD fall into two main categories:

- **The server does not hold any model and is only used as an aggregator** FedMD (Li and Wang, 2019) adopts a labeled public dataset for transfer learning among clients to seek fast improvement across all participants. Cronus (Chang et al., 2019) combines the local private dataset and the pseudo-labeled public dataset jointly for local training, where the pseudo-labels are ensembled with more robust aggregation rules.

- **The server holds a central model that acts as the target for collaborative training** FedKD (Gong et al., 2022) adopts a privacy-preserving ensemble strategy on cross-domain unlabeled data for one-way and one-shot distillation of the central model. In addition to server-side distillation, DS-FL (Itahara et al., 2021) also performs client-side distillation using the ensemble predictions on the unlabeled public dataset. Instead of transferring an ensemble of predictions, MHAT (Hu et al., 2021) achieves information aggregation by directly using predictions from multiple clients to train the central model simultaneously. However, these methods are generally

subject to confirmation bias caused by transferring knowledge over unlabeled data, which greatly limits their performance.

# 3 Preliminaries

## 3.1 Problem Definition

Consider a federated training environment with $K$ clients, where the $k$-th client holds a labeled private dataset $\mathcal{D}^k = \{(x_i^k, y_i^k)\}_{i=1}^{|\mathcal{D}^k|}$ drawn from the same or distinct distribution, along with a homogeneous or heterogeneous local model $f^k$ parameterized by $\theta^k$. The goal is to train a central model $f$ parameterized by $\theta$ on the server, but without direct access to these private data.

## 3.2 Federated Learning for NLP

In a general FL framework, the training process is divided into $T$ communication rounds through a server-client paradigm, where all clients share the same model architecture coordinated by a central server. Specifically, at the beginning of federated training, the server initializes the global model parameters $\theta_0$. At each communication round $t$, the training is proceeded as follows:

- **Broadcast** A subset of the client population $\mathcal{C}_t \subseteq \{1, 2, ..., K\}$ is sampled to participate in training, where $|\mathcal{C}_t| = \varepsilon \cdot K$, and $\varepsilon$ is the sampling rate. Then the server distributes the current global model parameters $\theta_{t-1}$ to the participating clients.

- **Local training** Each participating client $k \in \mathcal{C}_t$ uses the received parameters to initialize its local model,

$$\theta_{t-1}^k \leftarrow \theta_{t-1}, \qquad (1)$$

and updates it several epochs with its own private data $\mathcal{D}^k$,

$$\theta_t^k \leftarrow \theta_{t-1}^k - \eta \nabla \mathcal{L}_{CE}(\mathcal{D}^k; \theta_{t-1}^k), \qquad (2)$$

where $\eta$ is the learning rate of the central model, and $\mathcal{L}_{CE}$ denotes the loss function, which is usually a categorical cross-entropy for classification tasks.

- **Upload** The updated local model parameters $\theta_t^k$ are sent back to the server.

- **Aggregation** The server collects and aggregates the parameters from clients to obtain the

global model parameters for the next round,

$$\theta_t \leftarrow \sum_{k \in \mathcal{C}_t} \frac{|\mathcal{D}^k|}{|\mathcal{D}|} \theta_t^k, \qquad (3)$$

where $|\mathcal{D}^k|$ and $|\mathcal{D}| = \sum_{k \in \mathcal{C}_t} |\mathcal{D}^k|$ are the number of local data held by the $k$-th client and all participating clients, respectively.

## 3.3 Federated Distillation for NLP

In a general FD framework, an unlabeled public dataset $\mathcal{D}^0 = \{x_i^0\}_{i=1}^{|\mathcal{D}^0|}$ is hosted by the server and transmitted to all clients for knowledge transfer before the federated training starts. At each communication round $t$, the training process is summarized as the following steps:

- **Local training** Each participating client trains its local model $\theta_{t-1}^k$ on its own private data $\mathcal{D}^k$ for several epochs,

$$\theta_t^k \leftarrow \theta_{t-1}^k - \eta^k \nabla \mathcal{L}_{CE}(\mathcal{D}^k; \theta_{t-1}^k), \quad (4)$$

where $\eta^k$ is the learning rate of the $k$-th local model.

- **Local prediction** Each participating client computes its local predictions on the entire public proxy dataset $\mathcal{D}^0$ using its updated local model $\theta_t^k$,

$$Y_t^k = f^k(\mathcal{D}^0; \theta_t^k). \qquad (5)$$

- **Upload** Participating clients upload their local predictions to the server.

- **Aggregation** The predictions from clients are collected and aggregated by the server as ensemble predictions,

$$Y_t = \sum_{k \in \mathcal{C}_t} \frac{|\mathcal{D}^k|}{|\mathcal{D}|} Y_t^k. \qquad (6)$$

- **Server distillation** The ensemble predictions are treated as teacher knowledge to train the central model for several epochs,

$$\theta_t \leftarrow \theta_{t-1} - \eta \nabla \mathcal{L}_{CE}(Y_t, f(\mathcal{D}^0; \theta_{t-1})). \quad (7)$$

- **Broadcast** The server broadcasts the ensemble predictions to participating clients.

- **Local distillation** Each participating client distills its local model using the received ensemble predictions on the entire public proxy dataset,

$$\theta_t^k \leftarrow \theta_t^k - \eta^k \nabla \mathcal{L}_{CE}(Y_t, f^k(\mathcal{D}^0; \theta_t^k)). \quad (8)$$

## 4 Federated Interactive Distillation

In existing FD approaches, the central model is only allowed to passively mimic the local models by one-way knowledge transfer, leading to confirmation bias that heavily fades the superiority of FD. Instead of directly transmitting the entire public dataset and its predictions between the server and clients, the proposed FedID slices the unlabeled public dataset into multiple smaller batches for training, and handles only a small batch of data and predictions in each communication, which allows for an interaction between the central model and local models during the knowledge transfer process, while significantly reducing the load of a single communication. After each server distillation, the central model is allowed to feedback its performance on a small amount of labeled data held by the server back to each client to adapt its local model accordingly for rectifying its confirmation bias. The overall framework of FedID is presented in Figure 1.

### 4.1 Server Interactive Distillation

The server samples a batch of unlabeled public data $\mathbf{x}^0$ from $\mathcal{D}^0$ and distributes them to each participating client for local prediction,

$$\mathbf{y}_t^k = f^k(\mathbf{x}^0; \theta_t^k). \qquad (9)$$

The predictions from clients are uploaded to the server and aggregated with the same strategy as in Eq. (6),

$$\mathbf{y}_t = \sum_{k \in \mathcal{C}_t} \frac{|\mathcal{D}^k|}{|\mathcal{D}|} \mathbf{y}_t^k, \qquad (10)$$

together with the batch input $\mathbf{x}^0$, which are adopted to train the central model for knowledge transfer,

$$\theta_t \leftarrow \theta_{t-1} - \eta \nabla \mathcal{L}_{CE}(\mathbf{y}_t, f(\mathbf{x}^0; \theta_{t-1})). \quad (11)$$

The updated central model $\theta_t$ is then evaluated on a batch of data $(\mathbf{x}^{val}, \mathbf{y}^{val})$ sampled from the labeled dataset $\mathcal{D}^{val} = \{\mathbf{x}_i^{val}, \mathbf{y}_i^{val}\}_{i=1}^{|\mathcal{D}^{val}|}$ held by the server,

$$\begin{aligned} &\mathcal{L}_{CE}(\mathbf{y}^{val}, f(\mathbf{x}^{val}; \theta_t)) \\ &\triangleq \mathcal{L}_{CE}(\mathbf{y}^{val}, f(\mathbf{x}^{val}; \theta_{t-1} - \\ &\eta \nabla \mathcal{L}_{CE}(\sum_{k \in \mathcal{C}_t} \frac{|\mathcal{D}^k|}{|\mathcal{D}|} f^k(\mathbf{x}^0; \theta_t^k), f(\mathbf{x}^0; \theta_{t-1})))). \end{aligned} \qquad (12)$$

In addition to the ensemble predictions $\mathbf{y}_t$, the above-computed validation loss is also broadcast together to each participating client as feedback.

**Algorithm 1:** Federated Interactive Distillation (FedID)

---

**Input:** labeled private datasets $\{\mathcal{D}^k\}_{k=1}^K$; unlabeled public dataset $\mathcal{D}^0$; a handful of labeled dataset $\mathcal{D}^{val}$ held by the server; local models $\{\theta^k\}_{k=1}^K$; central model $\theta$; communication rounds $T$

**Output:** decentrally trained $\theta$

1 Each client initializes the local model $\theta_0^k$
2 Server initializes the central model $\theta_0$
3 **for** each communication round $t = 1, 2, ..., T$ **do**
4     $m \leftarrow \max(\varepsilon \cdot K, 1)$
5     $\mathcal{C}_t \leftarrow$ randomly sample a subset of $m$ clients from $K$ clients
6     **for** each client $k \in \mathcal{C}_t$ in parallel **do**
7         Update the local model parameters $\theta_t^k$ via Eq. (4)
8     **end**
9     **for** each mini-batch of unlabeled data $\mathbf{x}^0 \sim \mathcal{D}^0$ **do**
10         Server distributes a mini-batch of unlabeled data $\mathbf{x}^0$ to all participants $\mathcal{C}_t$
11         **for** each client $k \in \mathcal{C}_t$ in parallel **do**
12             Compute local predictions $\mathbf{y}_t^k$ on $\mathbf{x}^0$ via Eq. (9)
13             Upload local predictions $\mathbf{y}_t^k$ to the server
14         **end**
15         Server aggregates local predictions to create the ensemble predictions $\mathbf{y}_t$ via Eq. (10)
16         Server updates the central model parameters $\theta_t$ via Eq. (11)
17         Server samples a mini-batch of labeled data $(\mathbf{x}^{val}, \mathbf{y}^{val}) \sim \mathcal{D}^{val}$
18         Server computes the validation loss $\mathcal{L}_{CE}(\mathbf{y}^{val}, f(\mathbf{x}^{val}; \theta_t))$ via Eq. (12)
19         Server broadcasts the validation loss $\mathcal{L}_{CE}(\mathbf{y}^{val}, f(\mathbf{x}^{val}; \theta_t))$ and ensemble predictions $\mathbf{y}_t$ to all participants $\mathcal{C}_t$
20         **for** each client $k \in \mathcal{C}_t$ in parallel **do**
21             Update the local model parameters $\theta_t^k$ via Eq. (15)
22         **end**
23     **end**
24 **end**
25 **return** $\theta_T$

---

## 4.2 Client Interactive Distillation

For each participating client $k \in \mathcal{C}_t$, the gradients on the ensemble predictions $\mathbf{y}_t$ are computed to learn knowledge from other clients for alleviating data heterogeneity,

$$g_{distill}^k = \nabla_{\theta_t^k} \mathcal{L}_{CE}(\mathbf{y}_t, f^k(\mathbf{x}^0; \theta_t^k)). \qquad (13)$$

Also, the feedback gradients from the server to the client are computed from the validation loss,

$$g_{feedback}^k = \nabla_{\theta_t^k} \mathcal{L}_{CE}(\mathbf{y}^{val}, f(\mathbf{x}^{val}; \theta_t)), \quad (14)$$

and are added to further rectify its local model,

$$\theta_t^k \leftarrow \theta_t^k - \eta^k(g_{distill}^k + g_{feedback}^k). \qquad (15)$$

In this way, FedID establishes interactive distillation between the server and clients, where the client-to-server interaction aims to transfer the knowledge learned by local models during local training on their respective private data to the central model, while the server-to-client interaction attempts to rectify confirmation bias by allowing the local models to learn from the central model's feedback. The detailed procedures are summarized in Algorithms 1.

## 5 Experiments

### 5.1 Datasets

Considering that simulating data distributions with varying heterogeneity requires sampling by labels, we exclude the regression task STS-B (Cer et al., 2017) from the GLUE benchmark (Wang et al., 2019) because it lacks available labels for sampling, and take the remaining eight classification tasks for evaluation, including WNLI (Levesque et al., 2012), RTE (Dagan et al., 2005; Bar-Haim

| Method | WNLI (0.6k) Acc | RTE (2.5k) Acc | MRPC (3.7k) F1/Acc | CoLA (8.6k) Mcc | SST-2 (67k) Acc | QNLI (105k) Acc | QQP (364k) F1/Acc | MNLI (393k) Acc m/mm |
|---|---|---|---|---|---|---|---|---|
| Centralized | 56.3 | 61.4 | 85.9/79.2 | 52.4 | 92.4 | 90.2 | 86.3/89.7 | 82.8/83.5 |
| $\alpha$=100 | | | | | | | | |
| FedAvg | 49.6 | 56.7 | 84.5/78.0 | 51.0 | 91.8 | 89.8 | 85.5/88.2 | 82.0/82.7 |
| FedDF | 50.2 | 56.9 | 84.6/77.8 | 51.4 | **92.0** | 89.6 | **85.7/88.5** | 82.2/83.0 |
| FedED | 47.9 | 55.4 | 82.2/75.3 | 50.5 | 91.1 | 88.9 | 84.8/87.9 | 81.4/81.9 |
| FedKD | 45.4 | 53.6 | 80.3/74.0 | 48.3 | 89.2 | 86.1 | 82.6/86.0 | 79.8/80.1 |
| DS-FL | 50.6 | 56.4 | 84.0/78.4 | 51.7 | 90.7 | 89.1 | 84.9/88.2 | 82.6/83.4 |
| MHAT | 50.0 | 56.5 | 84.2/78.6 | 51.7 | 91.0 | 89.3 | 84.3/88.1 | 82.0/83.0 |
| FedID | **51.1** | **57.0** | **84.9/79.0** | **52.0** | 91.6 | **89.9** | 85.6/88.5 | **82.3/83.2** |
| $\alpha$=1 | | | | | | | | |
| FedAvg | 48.3 | 55.1 | 82.9/77.2 | 48.9 | 91.0 | 88.7 | 84.2/87.6 | 81.4/81.8 |
| FedDF | 50.2 | 56.4 | 83.2/77.3 | **49.8** | **91.2** | **89.0** | 84.5/88.0 | 81.8/82.1 |
| FedED | 46.8 | 55.0 | 82.0/76.4 | 47.8 | 90.7 | 88.1 | 83.1/87.5 | 80.8/81.3 |
| FedKD | 44.4 | 52.6 | 80.4/74.8 | 46.4 | 88.9 | 86.5 | 82.4/86.6 | 78.8/79.4 |
| DS-FL | 50.1 | 56.7 | 82.8/76.7 | 49.4 | 90.6 | 88.5 | 83.8/87.8 | 81.2/81.8 |
| MHAT | 50.2 | 56.7 | 82.6/76.8 | 49.2 | 90.7 | 88.8 | 84.1/87.4 | 81.6/82.0 |
| FedID | **50.9** | 56.7 | **83.2/77.3** | 49.6 | 90.9 | 88.8 | **84.7/88.1** | **81.9/82.3** |

Table 1: Experiment results of the homogeneous setting on the GLUE dev sets.

et al., 2006; Giampiccolo et al., 2007; Bentivogli et al., 2009), MRPC (Dolan and Brockett, 2005), CoLA (Warstadt et al., 2019), SST-2 (Socher et al., 2013), QNLI (Rajpurkar et al., 2016), QQP[1], and MNLI (Williams et al., 2018). See Appendix A for more details about GLUE.

For each task, the original development set is employed to evaluate the performance of the central and local models, while the original training set is divided into private and public datasets at a ratio of 1:1, which are used for client training and knowledge transfer between the server and clients, respectively. Particularly, for the resulting public dataset, we further sample 10% of it as the labeled dataset reserved for the server, and the rest as the unlabeled public dataset after rounding off labels.

Furthermore, to create disjoint client training data from the private dataset, the training instances of each client are drawn independently with class labels following a categorical distribution over $N$ classes parameterized by a vector $\mathbf{q}$ ($q_i \geq 0$, $i \in [1, N]$, and $\|\mathbf{q}\|_1 = 1$). Meanwhile, to simulate varying data distributions for clients, we further draw $\mathbf{q} \sim \text{Dir}(\alpha\mathbf{p})$ from a Dirichlet distribu-

tion (Hsu et al., 2019), where $\mathbf{p}$ is a prior class distribution over $N$ classes, and $\alpha$ is a concentration parameter that controls the degree of data heterogeneity among clients. Typically, when $\alpha \rightarrow \infty$, clients tend to be assigned to the identical data distribution, and conversely, when $\alpha \rightarrow 0$, clients are more likely to hold examples from only one random class. In our experiments, we set $\alpha$ to 100 and 1 to generate IID and non-IID data, respectively.[2]

### 5.2 Settings

**Homogeneous setting** For a homogeneous federated scenario, the model architectures of clients are limited to be the same as that of the server. To be compatible with FL methods for comparison, we adopt BERT-base (Devlin et al., 2019) as the central model since FL cannot usually be applied to larger PLMs due to communication bottlenecks.

**Heterogeneous setting** For a heterogeneous federated scenario, the central model is initialized with BERT-base, while each local model is selected from BERT-base, BERT-large, RoBERTa-base (Liu et al., 2019), or RoBERTa-large[3].

---

[1] https://quoradata.quora.com

[2] If not specified, $\alpha = 1$ is used by default.

[3] If not specified, heterogeneous setting is used by default.

| Method | WNLI (0.6k) Acc | RTE (2.5k) Acc | MRPC (3.7k) F1/Acc | CoLA (8.6k) Mcc | SST-2 (67k) Acc | QNLI (105k) Acc | QQP (364k) F1/Acc | MNLI (393k) Acc m/mm |
|---|---|---|---|---|---|---|---|---|
| \multicolumn{9}{c}{$\alpha=100$} | | | | | | | | |
| FedKD | 54.1 | 60.5 | 85.2/80.5 | 53.6 | 90.8 | 87.7 | 84.2/87.2 | 81.9/82.3 |
| DS-FL | 56.8 | 63.4 | 87.7/82.4 | 55.4 | 92.1 | 89.9 | 86.5/89.4 | 83.6/84.0 |
| MHAT | 56.9 | 63.2 | 87.7/82.5 | 55.7 | 92.0 | 90.6 | 87.0/89.5 | 84.0/84.2 |
| FedID | **58.2** | **64.6** | **88.5/83.6** | **56.5** | **92.7** | **91.4** | **88.1/91.2** | **84.6/84.6** |
| \multicolumn{9}{c}{$\alpha=1$} | | | | | | | | |
| FedKD | 52.7 | 58.4 | 84.3/80.0 | 50.0 | 88.9 | 87.1 | 84.2/87.5 | 81.2/81.6 |
| DS-FL | 55.6 | 60.3 | 86.6/81.6 | 53.9 | 90.9 | 89.8 | 86.4/89.1 | 83.0/83.5 |
| MHAT | 55.9 | 60.8 | 86.6/81.5 | 53.8 | 91.2 | 90.3 | 86.6/89.3 | 83.4/83.7 |
| FedID | **57.0** | **61.5** | **87.7/82.2** | **54.9** | **91.8** | **91.0** | **87.6/90.5** | **84.2/84.2** |

Table 2: Experiment results of the heterogeneous setting on the GLUE dev sets.

## 5.3 Implementation Details

We adopt the AdamW optimizer (Loshchilov and Hutter, 2019) with an initial learning rate of $2e$-5 to update the model parameters. For single-sentence or sentence-pair input to the model, the maximum sequence length is set to 128, and the batch size is set to 32. For hyperparameters in federated training, the number of epochs for local training, local distillation, and server distillation is set to 3, 3, and 3, respectively, the number of clients $K$ is set to 10, the fraction of client sampling $\varepsilon$ is set to 1, and the number of communication rounds $T$ is set to 10.

## 5.4 Baselines

We compare FedID with FL algorithms including FedAvg (Mcmahan et al., 2017), FedDF (Lin et al.), and FedED (Sui et al., 2020), as well as FD algorithms including FedKD (Gong et al., 2022), MAHT (Hu et al., 2021), and DS-FL (Itahara et al., 2021). We also provide the models with centralized training (denoted as Centralized) that have access to all private data held by the clients as an upper bound on model performance.[4]

## 5.5 Results

**Homogeneous setting**    Table 1 shows the performances across models in homogeneous setting. Without considering data privacy, centralized models always exhibit the best performance, while decentralized models sacrifice performance in ex-

---

[4]FedED and MHAT require a labeled public dataset in the original paper. For a fair comparison, a version using the unlabeled public dataset is provided in our implementation.

change for better privacy protection. However, this performance gap is gradually alleviated as the training data increases. In addition, the performances of FL and FD models are significantly degenerated when encountering non-IID data. Also, when sufficient public data is made available, the performance of FD models can be comparable to that of FL models, accompanied by lower communication costs.

**Heterogeneous setting**    Table 2 shows the performances across models in heterogeneous setting. The proposed FedID outperforms other baselines, demonstrating the superiority of tackling the confirmation bias. In particular, FedID exhibits strong robustness when only a small amount of training data is available, as there is not enough private data to adequately train the local models and thus the confirmation bias becomes more pronounced.

**Cross-domain setting**    We also use the original training sets of IMDB (Maas et al., 2011) and PAWS (Zhang et al., 2019) as unlabeled public data for SST-2 and QQP, respectively, to construct cross-domain knowledge transfer environments, where the confirmation bias is more likely to occur. The experimental results on the dev sets of SST-2 and QQP are shown in Table 3, where the greater performance gap between FedID and other baselines further confirms our claim.

## 5.6 Ablation Study

We remove the feedback gradient and the knowledge transfer gradient from Eq. (15), respectively,

| Method | SST-2 $\rightarrow$ IMDB Acc | QQP $\rightarrow$ PAWS F1/Acc |
|---|---|---|
| FedKD | 81.9 | 80.0/83.5 |
| DS-FL | 83.3 | 82.3/85.4 |
| MHAT | 83.9 | 82.8/85.6 |
| FedID | **86.4** | **84.1/87.1** |

Table 3: Results of models in the cross-domain setting.

| Method | RTE Acc | SST-2 Acc | QQP F1/Acc |
|---|---|---|---|
| FedID | 61.5 | 91.8 | 87.6/90.5 |
| w/o $g_{feedback}^k$ | 60.1 | 90.7 | 86.5/89.4 |
| w/o $g_{distill}^k$ | 60.6 | 91.1 | 86.9/89.8 |

Table 4: Results of model ablations on RTE, SST-2, and QQP.

| Method | Formulation |
|---|---|
| FedAvg | $(|\theta| \times (K+1)) \times T$ |
| FedDF | $(|\theta| \times (K+1) + |\mathcal{D}^0| \times K) \times T$ |
| FedED | $(|\theta| \times 1 + |\mathcal{D}^0| \times K) \times T$ |
| FedKD | $(|\mathcal{D}^0| \times K) \times 1$ |
| DS-FL | $(|\mathcal{D}^0| \times (K+1)) \times T$ |
| MHAT | $(|\mathcal{D}^0| \times (K+1)) \times T$ |
| FedID | $(|\mathcal{D}^0| \times (K+1)) \times T$ |

Table 5: Formulations of communication costs.

to conduct ablation experiments on the small dataset RTE, the medium dataset SST-2, and the large dataset QQP. As shown in Table 4, without the feedback gradient or the knowledge transfer gradient, the performances of models get worse, where the feedback gradient contributes more.

## 5.7 Communication Cost

Communication costs between the server and client models across baselines are presented in Table 5. The communication costs of FedAvg, FedDF, and FedED are much higher than those of FedKD, DS-FL, MHAT, and FedID as they entail extensive communication to share the model parameters. FedKD exhibits the lowest communication costs as the clients' predictions are aggregated without the need to send back to the clients and only one round of communication is executed, while DS-FL and MHAT need to broadcast the ensemble predictions from the server back to each client. Similarly, FedID is required to transmit ensemble predictions and the validation loss in batches to clients as feedback, but the communication costs for the validation loss are negligible compared to that of ensemble predictions. As a result, the communication costs of FedID remain in line with DS-FL and MHAT, but the communication between the server and clients is more frequent.

## 5.8 Effect of Unlabeled Public Dataset Size

In our experimental setup, we partition the original training dataset into a private dataset and a public dataset. To further investigate the effect of different proportions of the public dataset on performance, we keep the size of the private dataset constant while conducting experiments using 10%, 20%, 40%, 80%, and 100% of the public dataset, respectively. The results in Figure 2 show that the performance of the central model improves to some extent as the size of the public dataset increases, where FedID still exhibits superior performance and robustness.

## 5.9 Effect of Labeled Dataset Size

To investigate the effect of the size of the labeled dataset retained by the server on performance, we experiment with 10%, 20%, 40%, 80%, and 100% of the labeled dataset, respectively. For FedID, the labeled data is used to rectify the confirmation bias in the client models' predictions, while for other FD methods, the labeled data is added to the training of the central model. The results in Figure 3 show that FedID is least sensitive to the size of the labeled dataset since this data is not used to directly participate in the training of the central model. Moreover, although other FD methods use the labeled data directly for additional training of the central model, there is still no significant performance improvement observed because the proportion of the labeled data is far lower than that of the unlabeled data, and thus its supervision on the central model is limited. Our solution makes better use of the small amount of labeled data by leveraging it to rectify confirmation bias in the predictions from unlabeled data.

## 5.10 Effect of Number of Clients

The number of clients usually imposes a significant impact on performance, as the entire training dataset is partitioned and distributed to multiple clients. To investigate this, we increase the number of clients from 5 to 10 and 20 while keeping

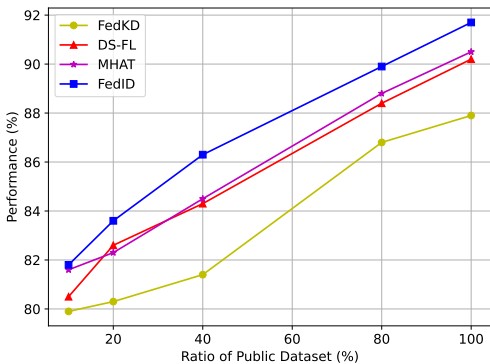

Figure 2: Performance of the central model on SST-2 with different sizes of public dataset.

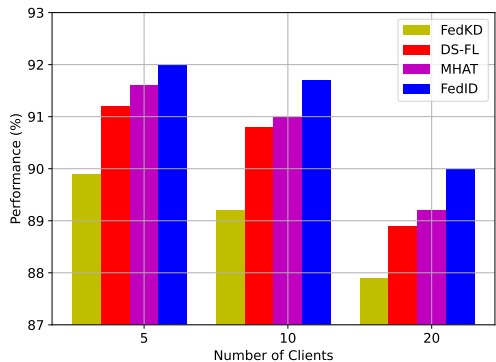

Figure 4: Performance of the central model on SST-2 with different numbers of clients.

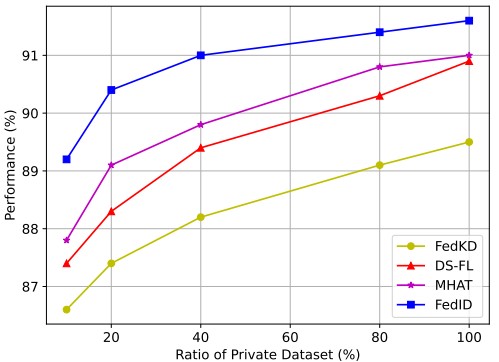

Figure 3: Performance of the central model on SST-2 with different sizes of labeled dataset.

the total amount of private data for all clients constant, which results in a corresponding change in the quantity of local private data that can be allocated to each client. The results depicted in Figure 4 show a decrease in the performance of the central model as the number of clients increases, which can be attributed to the fact that the reduction in local private data makes it challenging to achieve adequate local training. However, FedID shows better robustness in response to this change due to the mitigation of confirmation bias.

## 6 Conclusions

This study explores the application of FD to decentralized training of large-scale PLMs in homogeneous and heterogeneous settings, and further presents an interactive FD scheme to mitigate the confirmation bias caused by transferring knowledge on an unlabeled public dataset. Moreover, a benchmarking framework across multiple tasks with diverse data distributions is developed to contribute to the research of FD in NLP community. Future work will be executed to aggregate differ-

entially private local predictions for a stronger privacy guarantee, enhancing the resilience of FedID against malicious server or clients.

## Limitations

There are two main limitations to our work compared to previous efforts: 1) We assume that a small amount of labeled data is retained in the server. However, this situation may be common in real life. For instance, an institution possesses only a small amount of training data, which is not enough to train a well-performing model, thus it may want to resort to collaborative training with other institutions with the help of FD on a large amount of unlabeled public data. However, directly transferring knowledge on the unlabeled data may not yield a satisfactory performance, while the small amount of training data retained by the institution can be used as labeled data by the proposed FedID to maximize the performance. In addition, our approach is more suitable for the case where one client in the federation acts as the server; 2) Compared with other FD approaches, our solution slices the unlabeled public dataset into multiple smaller batches for training, thus entailing more frequent communication between the server and clients. However, the increase in communication frequency may be tolerable considering the similar communication costs and the fact that transmitting smaller packets avoids potential network congestion when the public dataset is too large.

## Ethics Statement

This study aims to explore an alternative decentralized training paradigm for large-scale PLMs, and the proposed method does pose ethical issues or potential biases. All models, baselines, and datasets

used in this work are publicly available and widely used.

## Acknowledgements

This work was supported by the National Natural Science Foundation of China (NSFC) under Grant Nos.61966038 and 62266051, and the Postgraduate Research and Innovation Foundation of Yunnan University under Grant No.KC-23236531. The authors would like to thank the anonymous reviewers for their constructive comments.

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

## A  Details of GLUE Benchmark

GLUE (Wang et al., 2019) is a benchmarking framework designed to assess the performance and generalization capability of NLP models especially large-scale PLMs across nine NLP tasks. The descriptions of each task are presented as follows:

- **WNLI** The Winograd Natural Language Inference (Levesque et al., 2012) is a sentence-pair binary classification task that requires the model to determine whether two sentences in a given sentence-pair are entailment relations, with the evaluation metric of accuracy.

- **RTE** The Recognizing Textual Entailment (Dagan et al., 2005; Bar-Haim et al., 2006; Giampiccolo et al., 2007; Bentivogli et al., 2009) is a sentence-pair binary classification task, which requires the model to determine whether two sentences in a given sentence pair are entailment relations, with the evaluation metric of accuracy.

- **MRPC** The Microsoft Research Paraphrase Corpus (Dolan and Brockett, 2005) is a sentence-pair binary classification task that requires the model to determine whether two sentences in a given sentence pair are semantically equivalent, with evaluation metrics of accuracy and $F_1$-score.

- **STS-B** The Semantic Textual Similarity Benchmark (Cer et al., 2017) is a sentence-pair regression task that requires the model to evaluate how similar two sentences in a given sentence-pair are by a floating score range from 0 to 5, with evaluation metrics of Pearson and Spearman correlations.

- **CoLA** The Corpus of Linguistic Acceptability (Warstadt et al., 2019) is a single-sentence binary classification task that requires the

| Dataset | #Train | #Dev | #Test |
|---------|--------|------|-------|
| WNLI | 635 | 71 | 146 |
| RTE | 2490 | 277 | 3000 |
| MRPC | 3668 | 408 | 1725 |
| STS-B | 5749 | 1500 | 1379 |
| CoLA | 8551 | 1043 | 1063 |
| SST-2 | 67349 | 872 | 1821 |
| QNLI | 104743 | 5463 | 5463 |
| QQP | 363846 | 40430 | 390965 |
| MNLI-m | 392702 | 9815 | 9796 |
| MNLI-mm | | 9832 | 9847 |

Table 6: Statistics of the GLUE benchmark.

model to determine whether a given English sentence is grammatically correct, with the evaluation metric of the Matthews correlation.

- **SST-2** The Stanford Sentiment Treebank (Socher et al., 2013) is a single-sentence binary classification task that requires the model to determine whether a given movie review is positive or negative in sentiment, with the evaluation metric of accuracy.

- **QNLI** The Question Natural Language Inference (Rajpurkar et al., 2016) is a sentence-pair binary classification task. Given a question and a context, the model is required to determine whether the context contains the answer to the question, with the evaluation metric of accuracy.

- **QQP** The Quora Question Pairs is a sentence-pair binary classification task. Given a pair of questions, the model is required to determine whether the two sentences are semantically equivalent, with evaluation metrics of accuracy and $F_1$-score.

- **MNLI** The Multi-genre Natural Language Inference (Williams et al., 2018) is a sentence-pair three-way classification task. Given a premise and a hypothesis, the model is required to determine whether the hypothesis is an entailment, contradiction, or neutral with respect to the premise. The task is divided into matched and mismatched versions, with evaluation metrics of matched accuracy and mismatched accuracy, respectively.

The statistics of these tasks are presented in Table 6.