# OpenReview forum: "FedID: Federated Interactive Distillation for Large-Scale Pretraining Language Models"
_EMNLP/2023/Conference — EMNLP 2023 Main_

### Official Review · Reviewer_VAtG · 2023-08-03

**Soundness:** 5

**Excitement:**

4: Strong: This paper deepens the understanding of some phenomenon or lowers the barriers to an existing research direction.

**Paper Topic And Main Contributions:**

The paper proposes a new training algorithm in the federated distillation setting. The proposed approach uses a public proxy dataset for collecting client knowledge and reflects the gradient of the small labeled dataset to mitigate the problem of confirmation bias. Experiments on various GLUE tasks show that the proposed method is effective and outperform many baselines while being robust.

**Questions For The Authors:**

1. Are there any design intuitions behind the interaction procedure? Mentioning the intuition in the introduction could help readers better understand the paper.
2. With various types of tasks in GLUE being evaluated, are there any insights in the influence of task type in whether interactive distillation is useful?


**Reasons To Accept:**

1. Useful privacy-preserving training method with good performance
2. Reasonable and effective algorithm design
3. Comprehensive experiment to show the effectiveness of the method


**Reasons To Reject:**

1. Lack case-level comparison to analyze/justify in detail why the proposed method alleviates confirmation bias
2. Lack of comparison with alternative designs (e.g. for server-to-client interaction, can we also use the proxy dataset instead of the gradient?)
3. Lack real-world use case analysis for the proposed training scenario (in the limitation section the author explains a constructed scenario which is reasonable, but still it would be informative to find concrete use case)


**Reproducibility:**

4: Could mostly reproduce the results, but there may be some variation because of sample variance or minor variations in their interpretation of the protocol or method.

**Reviewer Confidence:**

4: Quite sure. I tried to check the important points carefully. It's unlikely, though conceivable, that I missed something that should affect my ratings.

**Typos Grammar Style And Presentation Improvements:**

The introduction does not provide enough context about FD to understand the general contribution. The authors may consider adding a summarized version of section 2.2 to the introduction to help readers better understand what is being investigated in this paper.
Table 4: feedback should be subscripted.

---

> ### Author Rebuttal · Authors · 2023-08-28
>
> We deeply value your insightful comments and suggestions. Each comment has significantly enhanced the quality of the paper.
>
> **Reasons To Reject 1:** Lack case-level comparison to analyze/justify in detail why the proposed method alleviates confirmation bias.
>
> **Response:** We will add a case study section to verify that the proposed method alleviates confirmation bias by calculating whether local model predictions are more accurate after each interaction than when there is no interaction.
>
> **Reasons To Reject 2:** Lack of comparison with alternative designs (e.g., for server-to-client interaction, can we also use the proxy dataset instead of the gradient?)
>
> **Response:** We will rich the ablation study in Section 5.6, to examine the superiority of using the feedback gradient compared to the proxy dataset in mitigating confirmation bias.
>
> **Reasons To Reject 3:** Lack real-world use case analysis for the proposed training scenario (in the limitation section the author explains a constructed scenario which is reasonable, but still it would be informative to find concrete use case)
>
> **Response:** The proposed FedID can be applied to many real-world cases. For example, an accurate medical text classification system requires the proposed training scenario, to quickly predict which category of disease a patient belongs to based on his/her symptoms for assigning the appropriate departments and specialists in a timely and convenient manner. Specifically, a hospital usually has only a small amount of training data, which is not enough to train a well-performing model, thus it may want to resort to the private data and models from other hospitals with the help of FD on a large amount of unlabeled public medical data. The small amount of training data retained by the hospital can be used as labeled data by the proposed FedID to maximize the performance. We will elaborate on this use case analysis in the limitation section.
>
> **Questions For The Authors 1:** Are there any design intuitions behind the interaction procedure? Mentioning the intuition in the introduction could help readers better understand the paper.
>
> **Response:** Yes. In existing FD approaches, the central model only can passively mimic the local models, which results in the confirmation bias that heavily fades the superiority of FD. Different from the existing approaches, the proposed FedID establishes a link between the central model and the local models by introducing an interaction mechanism. This interaction allows the central model to provide timely performance feedback to adjust the local models for progressively correcting confirmation bias, thereby improving performance. We will also add this intuition to the introduction to assist readers in understanding our work.
>
> **Questions For The Authors 2:** With various types of tasks in GLUE being evaluated, are there any insights in the influence of task type in whether interactive distillation is useful?
>
> **Response:** Yes, Tables 1 and 2 show that interactive distillation has subtle discrepancies across tasks. Overall, for tasks with more data, the effect of interaction distillation will be smaller because the client has enough data for local training, in which the problem of confirmation bias will be less obvious. For tasks with more labels, interactive distillation is more useful because more labels mean that confirmation bias is more likely to occur. We will also add these findings to Section 5.5 to provide more interpretation for the experimental results.
>
> **Typos Grammar Style And Presentation Improvements 1:** The introduction does not provide enough context about FD to understand the general contribution. The authors may consider adding a summarized version of section 2.2 to the introduction to help readers better understand what is being investigated in this paper.
>
> **Response:** We will add more details in the introduction to provide readers with more preliminaries about federal distillation as well as its recent efforts and limitations.
>
> **Typos Grammar Style And Presentation Improvements 2:** Table 4: feedback should be subscripted.
>
> **Response:** We have rechecked the paper and corrected these typos.

---

### Official Review · Reviewer_aUk1 · 2023-08-03

**Soundness:** 4

**Excitement:**

4: Strong: This paper deepens the understanding of some phenomenon or lowers the barriers to an existing research direction.

**Missing References:**

Please consider:
N. Papernot, M. Abadi, U. Erlingsson, I. Goodfellow, and K. Talwar. Semisupervised Knowledge Transfer for Deep Learning from Private Training Data. arXiv, abs/1610.05755, 2017.
in the context of FD where the authors show how multiple (teacher) nodes trained on sensitive data, can further train a (student) model based on differentially private aggregated outputs of the teacher nodes.

**Paper Topic And Main Contributions:**

The authors propose a Federated Learning variant - called Federated Interactive Distillation (FedID) - for Pre-trained Language Models (PLMs). They then validate its performance with extensive experimental results on GLUE tasks for both homogeneous and heterogeneous model settings.

**Questions For The Authors:**

Given the interactive nature of FedID, it leads to a higher communication overhead - so please elaborate on the below statement:
"but the communication cost of validation loss is negligible compared to that of ensemble predictions."
It seems that the authors are focusing primarily on the transmitted data size, and not the communication rounds. Please elaborate.

On what basis were the 8 GLUE classification tasks selected?

**Reasons To Accept:**

The FedID approach is generic and can potentially be applied to train non-NLP models as well in a Federated Learning setting. In this paper, the authors specifically apply it to PLMs and show their efficiency and performance benefits for NLP tasks - making it relevant for EMNLP.

The experimental results are comprehensive providing interesting insights around public dataset distribution between clients and communication aspects. The heterogenous setting is of course more interesting from a 'confirmation bias' perspective, where the authors show FedID's robustness when only a small amount of training data is available that are insufficient to train the local models.

**Reasons To Reject:**

FedID is more of an incremental contribution over existing Federated Distillation (FD) approaches, with a different distribution mechanism. So the strength of paper is primarily on the applied part.

For Federated Learning contributions, it is always good to assess how resilient it is to malicious clients/server. So I would encourage the authors to add this discussion.

**Reproducibility:**

4: Could mostly reproduce the results, but there may be some variation because of sample variance or minor variations in their interpretation of the protocol or method.

**Reviewer Confidence:**

4: Quite sure. I tried to check the important points carefully. It's unlikely, though conceivable, that I missed something that should affect my ratings.

---

> ### Author Rebuttal · Authors · 2023-08-28
>
> We greatly value your perceptive remarks and recommendations, which contribute a lot to improving the quality of the paper.
>
> **Reasons To Reject 1:** FedID is more of an incremental contribution over existing Federated Distillation (FD) approaches, with a different distribution mechanism. So the strength of paper is primarily on the applied part.
>
> **Response:** In existing FD approaches, the central model can only passively mimic the local models, which results in the confirmation bias that heavily fades the superiority of FD. Different from the existing approaches, the proposed FedID establishes a link between the central model and the local models by introducing an interaction mechanism. This interaction allows the central model to provide timely performance feedback to adjust the local models for progressively correcting confirmation bias, thereby improving performance. We will highlight this contribution in the introduction section.
>
> **Reasons To Reject 2:** For Federated Learning contributions, it is always good to assess how resilient it is to malicious clients/server. So I would encourage the authors to add this discussion.
>
> **Response:** In the current version, we focused mainly on how to improve the performance of FD and therefore neglected to protect against malicious attacks from the server or clients. Thanks to your suggested references, we identify that FedID can be well integrated with the Private Aggregation of Teacher Ensemble (PATE) to compensate for this deficiency by aggregating differentially private local predictions for a stronger privacy guarantee. We will perform additional experiments about resilience to malicious attacks to demonstrate this improvement. Thanks again for your recommendation.
>
> **Questions For The Authors 1:** Given the interactive nature of FedID, it leads to a higher communication overhead – so please elaborate on the below statement: "but the communication cost of validation loss is negligible compared to that of ensemble predictions". It seems that the authors are focusing primarily on the transmitted data size, and not the communication rounds. Please elaborate.
>
> **Response:** We will reformulate the communication cost in Section 5.7 by considering the transmitted data size and the communication rounds together. As a result, the communication cost of FedID will higher due to more frequent communication between the server and clients. However, as we discussed in the limitations, the increase in communication frequency may be tolerable due to the fact that FedID transmits smaller packets avoiding potential network congestion if the public dataset is too large. This is another merit.
>
> **Questions For The Authors 2:** On what basis were the 8 GLUE classification tasks selected?
>
> **Response:** It is well known that GLUE contains 8 classification tasks and 1 regression task. Considering we use the Dirichlet distribution to simulate data distribution with different heterogeneity, which samples by label, thus we selected all classification tasks and excluded the regression task because it lacks available labels for sampling.

---

### Official Review · Reviewer_kBJp · 2023-08-12

**Soundness:** 3

**Excitement:**

3: Ambivalent: It has merits (e.g., it reports state-of-the-art results, the idea is nice), but there are key weaknesses (e.g., it describes incremental work), and it can significantly benefit from another round of revision. However, I won't object to accepting it if my co-reviewers champion it.

**Paper Topic And Main Contributions:**

The authors investigated FD in homogeneous and heterogeneous large-scale pre-trained language models and proposed a novel Federated Interactive Distillation to mitigate the problem of misleading privileged knowledge caused by confirmation bias in conventional FD.

**Questions For The Authors:**

There is no report on hyperparameter settings and code, how to ensure the reproducibility of the paper？

**Reasons To Accept:**

1. It is a good idea to solve the homogeneous and heterogeneous in federated distillation through some labeled data of the server
2. The author has done sufficient experiments to verify the effectiveness of the method

**Reasons To Reject:**

1. The homogeneous and heterogeneous of PLM and server are the homogeneous and heterogeneous faced by other models. It seems that there is not much difference. As the author said, when the server model is PLM, the data can be reasonably divided into public data on the server and private data on the client, but it can also be used in other FDs in visual scenes, recommendation scenes, and audio scenes. Therefore, the first exploration of the server model is that the homogeneous and heterogeneous problems in FD of PLM do not seem to be a contribution, so the contribution seems to be a bit weak.
2. Although many published federated learning papers lack theories, if there are some more important analyzes such as errors, it will make the papers more solid
3. Lack of any reproducible content, including code, experiment/hyperparameter settings,

**Reproducibility:**

3: Could reproduce the results with some difficulty. The settings of parameters are underspecified or subjectively determined; the training/evaluation data are not widely available.

**Reviewer Confidence:**

3: Pretty sure, but there's a chance I missed something. Although I have a good feel for this area in general, I did not carefully check the paper's details, e.g., the math, experimental design, or novelty.

---

> ### Author Rebuttal · Authors · 2023-08-28
>
> We highly appreciate your insightful comments and suggestions. Each comment has significantly improved the quality of the paper.
>
> **Reasons To Reject 1:** The homogeneous and heterogeneous problems in FD of PLM do not seem to be a contribution, so the contribution seems to be a bit weak.
>
> **Response:** Recent federated work in NLP only considers the decentralized training of homogeneous PLMs, but does not explore how the heterogeneous PLMs perform decentralized training. Therefore, we mainly focus on investigating FD in homogeneous and heterogeneous large-scale PLMs. Thanks again for your suggestion about the data heterogeneity on different modalities, our future work will explore heterogeneous federated learning in multi-modality settings.
>
> **Reasons To Reject 2:** Although many published federated learning papers lack theories, if there are some more important analyzes such as errors, it will make the papers more solid.
>
> **Response:** We will consolidate this paper with more perspectives on empirical risk minimization, convergence analysis, and generalization bounds.
>
> **Reasons To Reject 3:** Lack of any reproducible content, including code, experiment/hyperparameter settings.
>
> **Response:** In addition to the implementation details provided in Section 5.3, due to the page limitation, the codes and more experimental setups are illustrated in the "Supplementary Materials". If there are some spaces left, we will present more experimental settings in the final version. In addition, we will also make the codes publicly available.
>
> **Questions For The Authors 1:** There is no report on hyperparameter setting and code, how to ensure the reproducibility of the paper?
>
> **Response:** Please refer to **Reasons To Reject 3**.

---

### Meta-Review · Area_Chair_E5Du · 2023-09-12

**Recommendation:** 4

**Metareview:**

Thanks reviewers so much your efforts in providing comprehensive reviews and comments to improve the paper.

Thanks authors for providing actionable rebuttals and facilitating discussions.

Summary: The authors conducted research on Federated Distillation (FD) in large-scale pre-trained language models, both homogeneous and heterogeneous. They introduced Federated Interactive Distillation (FedID) to address confirmation bias issues in FD. FedID is a Federated Learning variant for Pre-trained Language Models (PLMs) and was tested extensively on GLUE tasks in various model settings. Their approach includes a new training algorithm that uses a public proxy dataset to collect client knowledge and adjusts the gradient of the small labeled dataset to combat confirmation bias. Experimental results demonstrated the effectiveness and robustness of their proposed method, which outperformed several baseline approaches in different GLUE tasks.

Average Soundness: (3+4+5)/3 = 4
Average Excitement: (3+4+4)/3 = 3.7
Reproducibility: (3+4+4)/3 = 3.7

Summary of Pros:
+ Solve the homogeneous and heterogeneous in federated distillation through some labeled data of the server
+ Comprehensive experiments providing interesting insights around public dataset distribution between clients and communication aspects
+ Relevant for EMNLP.
+ Useful privacy-preserving training method with good performance
+ Reasonable and effective algorithm design

Summary of Cons:
+ Lack theories
+ Lack of reproducible content
+ Contribution is incremental
+ Lack of discussion on resilience
+ Need more comparisons such as case-level analysis, alternative designs, real-world use case analysis

The authors have clarified and promised the changes to tackle the cons. The changes are feasible and can be timely ready for the final version.

---

### Decision · Program_Chairs · 2023-10-07

**Decision:**

Accept-Main

**Comment:**

Thanks reviewers so much your efforts in providing comprehensive reviews and comments to improve the paper.

Thanks authors for providing actionable rebuttals and facilitating discussions.

Summary: The authors conducted research on Federated Distillation (FD) in large-scale pre-trained language models, both homogeneous and heterogeneous. They introduced Federated Interactive Distillation (FedID) to address confirmation bias issues in FD. FedID is a Federated Learning variant for Pre-trained Language Models (PLMs) and was tested extensively on GLUE tasks in various model settings. Their approach includes a new training algorithm that uses a public proxy dataset to collect client knowledge and adjusts the gradient of the small labeled dataset to combat confirmation bias. Experimental results demonstrated the effectiveness and robustness of their proposed method, which outperformed several baseline approaches in different GLUE tasks.

Average Soundness: (3+4+5)/3 = 4
Average Excitement: (3+4+4)/3 = 3.7
Reproducibility: (3+4+4)/3 = 3.7

Summary of Pros:
+ Solve the homogeneous and heterogeneous in federated distillation through some labeled data of the server
+ Comprehensive experiments providing interesting insights around public dataset distribution between clients and communication aspects
+ Relevant for EMNLP.
+ Useful privacy-preserving training method with good performance
+ Reasonable and effective algorithm design

Summary of Cons:
+ Lack theories
+ Lack of reproducible content
+ Contribution is incremental
+ Lack of discussion on resilience
+ Need more comparisons such as case-level analysis, alternative designs, real-world use case analysis

The authors have clarified and promised the changes to tackle the cons. The changes are feasible and can be timely ready for the final version.